# Effect of a Remotely Delivered Weight Loss Intervention in Early-Stage Breast Cancer: Randomized Controlled Trial

**DOI:** 10.3390/nu13114091

**Published:** 2021-11-15

**Authors:** Marina M. Reeves, Caroline O. Terranova, Elisabeth A. H. Winkler, Nicole McCarthy, Ingrid J. Hickman, Robert S. Ware, Sheleigh P. Lawler, Elizabeth G. Eakin, Wendy Demark-Wahnefried

**Affiliations:** 1School of Public Health, The University of Queensland, Brisbane 4006, Australia; caroline.terranova@qut.edu.au (C.O.T.); e.winkler@sph.uq.edu.au (E.A.H.W.); s.lawler@sph.uq.edu.au (S.P.L.); e.eakin@uq.edu.au (E.G.E.); 2Wesley Centre, Icon Cancer Care, Brisbane 4066, Australia; Nicole.McCarthy@icon.team; 3Department of Nutrition and Dietetics, Princess Alexandra Hospital, Brisbane 4102, Australia; i.hickman@uq.edu.au; 4Menzies Health Institute Queensland, Griffith University, Brisbane 4111, Australia; r.ware@griffith.edu.au; 5O’Neal Comprehensive Cancer Center, University of Alabama at Birmingham, Birmingham, AL 35294, USA; demark@uab.edu

**Keywords:** obesity, exercise, nutrition, supportive care, survivorship, telehealth

## Abstract

Limited evidence exists on the effects of weight loss on chronic disease risk and patient-reported outcomes in breast cancer survivors. Breast cancer survivors (stage I–III; body mass index 25–45 kg/m^2^) were randomized to a 12-month, remotely delivered (22 telephone calls, mailed material, optional text messages) weight loss (diet and physical activity) intervention (*n* = 79) or usual care (*n* = 80). Weight loss (primary outcome), body composition, metabolic syndrome risk score and components, quality of life, fatigue, musculoskeletal pain, menopausal symptoms, fear of recurrence, and body image were assessed at baseline, 6 months, 12 months (primary endpoint), and 18 months. Participants were 55 ± 9 years and 10.7 ± 5.0 months post-diagnosis; retention was 81.8% (12 months) and 80.5% (18 months). At 12-months, intervention participants had significantly greater improvements in weight (−4.5% [95%CI: −6.5, −2.5]; *p* < 0.001), fat mass (−3.3 kg [−4.8, −1.9]; *p* < 0.001), metabolic syndrome risk score (−0.19 [−0.32, −0.05]; *p* = 0.006), waist circumference (−3.2 cm [−5.5, −0.9]; *p* = 0.007), fasting plasma glucose (−0.23 mmol/L [−0.44, −0.02]; *p* = 0.032), physical quality of life (2.7 [0.7, 4.6]; *p* = 0.007; Cohen’s effect size (*d*) = 0.40), musculoskeletal pain (−0.5 [−0.8, −0.2]; *p* = 0.003; *d* = 0.49), and body image (−0.2 [−0.4, −0.0]; *p* = 0.030; *d* = 0.31) than usual care. At 18 months, effects on weight, adiposity, and metabolic syndrome risk scores were sustained; however, significant reductions in lean mass were observed (−1.1 kg [−1.7, −0.4]; *p* < 0.001). This intervention led to sustained improvements in adiposity and metabolic syndrome risk.

## 1. Introduction

Attention has been focused on modifiable risk factors (diet, obesity, physical activity) as a means to improve breast cancer outcomes [1,2]. Physical activity has been associated with reduced breast cancer recurrence risk and increased survival [3,4], with exercise interventions producing improvements in quality of life, physical function, and fatigue [5,6,7]. Breast cancer survivors who maintain a healthful weight (body mass index (BMI) = 18.5–24.9 kg/m^2^) have 30–40% reduced mortality risk compared to those with obesity (BMI ≥ 30 kg/m^2^) [8]. Consequently, weight management, physical activity, and dietary changes are encouraged for breast cancer survivors [1,2,9,10].

Weight loss trials in early-stage breast cancer have shown that modest weight loss is safe and feasible [11,12], with ongoing trials assessing effects on survival [13,14,15]. With limited evidence on prognostic benefit, there remains a need to understand the broader effects of weight loss on outcomes such as body composition [16], chronic disease risk (given that cardiovascular disease deaths surpass cancer-specific mortality for the majority of breast cancer survivors) [17], and patient-reported outcomes, i.e., quality of life and treatment-related side effects. Treatment-related side effects such as fatigue, arthralgia, and menopausal symptoms can persist long-term [18] and are exacerbated by excess body weight [19,20,21,22,23]. With the exception of quality of life, few trials have examined the effect of weight loss on patient-reported outcomes [11,12]. Further, weight loss trials to date have only evaluated effects on individual metabolic biomarkers and not broader measures of chronic disease risk [11,12]. A recent exercise-only trial reported large improvements in metabolic syndrome risk, following a short-term, supervised exercise intervention [24]. Metabolic syndrome, a cluster of risk factors that increases cardiovascular disease and type 2 diabetes risk [25], has also been associated with increased breast cancer mortality and recurrence risk [26,27].

Importantly, of relevance in the current COVID-19 environment is the need to understand the benefits that can be achieved with remotely delivered interventions (no face-to-face contact). The ‘Living Well after Breast Cancer’ trial aimed to evaluate the effectiveness of a 12-month, remotely delivered weight loss intervention versus usual care in women following treatment for early-stage breast cancer. This paper reports on the effects of the intervention on the primary outcome (percent weight loss), body composition, metabolic syndrome risk, and patient-reported outcomes [28], including whether intervention effects were sustained 6 months after intervention completion.

## 2. Materials and Methods

This two-arm, parallel, randomized trial was registered with the Australian and New Zealand Clinical Trial Registry (ACTRN12612000997853), with the trial protocol previously published [28]. The human research ethics committees of the Royal Brisbane & Women’s Hospital, Greenslopes Private Hospital, St. Vincent’s Health & Aged Care, and the University of Queensland granted approval. Signed, informed consent was obtained prior to participation.

### 2.1. Participants and Recruitment

Participants were recruited from seven hospitals in Brisbane, Australia, and the state-based cancer registry between October 2012 and December 2014. Women aged 18–75 years were eligible if they had: a diagnosis of stage I–III breast cancer in the previous two years, a BMI 25–45 kg/m^2^, and completed primary cancer treatment (excluding endocrine treatment). Exclusions included pregnancy, contraindications to unsupervised exercise, >5% weight loss within the previous six months, insufficient English, or self-reported anxiety and/or depression that would interfere with participation [28]. Following baseline assessment, an off-site staff member randomized participants (1:1) into intervention or usual care arms using a computer-generated randomization program with uneven block sizes.

### 2.2. Usual Care

Participants in both arms received materials after each assessment, including a study newsletter and assessment feedback. Participants allocated to usual care received brief feedback on their assessment results, whereas for intervention participants, assessment results were compared to guidelines.

### 2.3. Weight Loss Intervention

The intervention was based on clinical practice guidelines for overweight and obesity (consistent with recommendations for cancer survivors) [9,10,29], piloted in a feasibility study [30,31], and described previously [28]. The intervention was remotely delivered via telephone by accredited dietitians (with optional text messages) and aimed for weight loss of 5–10%, by reducing energy intake (1200–1500 kcal/day) [32] and saturated fat (<7% total energy), increasing vegetables and fruit (5 and 2 servings/day, respectively), and limiting alcohol (≤1 serving/day). Additionally, incremental increases in moderate-to-vigorous intensity aerobic activity to 210 min/week and 2–3 resistance exercise sessions/week were encouraged. The intervention was grounded in social cognitive theory [33], emphasizing self-monitoring, goal setting, social support, problem solving, stimulus control, positive self-talk, and self-reward.

Intervention participants received a workbook, scale, measuring tape, pedometer, calorie-counter book, and self-monitoring diary. During the first 6 months, participants received up to 16 calls (six weekly then 10 bi-weekly calls) and optional text messages. During the second 6 months, participants received six monthly calls and tailored text messages. Dietitians used a semi-structured approach and motivational interviewing for each call.

### 2.4. Data Collection

Data were collected at baseline, 6 months, 12 months (primary endpoint), and 18 months by staff blinded to arm assignment. Methods and reliability/validity of measures have been reported previously [28].

#### 2.4.1. Primary Outcome

Weight was measured without heavy clothing or shoes to the nearest 0.1 kg (Tanita BWB−600 Wedderburn Scales, Sydney, Australia), in duplicate, with the mean used, and expressed as percent weight change from baseline.

#### 2.4.2. Secondary Outcomes

Secondary outcomes (all continuous) were body composition (total fat and lean mass), biomarkers of metabolic syndrome (risk score, waist circumference, triglycerides, high density lipoprotein (HDL) cholesterol, systolic and diastolic blood pressure, fasting plasma glucose), quality of life, fatigue, arthralgia, menopausal symptoms, fear of cancer recurrence, and body image. Detailed regional body composition outcomes were also explored.

Body composition was measured by a trained technician using Dual-Energy X-ray Absorptiometry (Lunar Prodigy, GE Medical Systems, Madison, WI, USA). Waist circumference was measured at the iliac crest in duplicate, with the mean used. Blood pressure was measured seated using an automated sphygmomanometer (300 Series Vital Signs Monitor, Welch Allyn, Beaverton, OR, USA) in duplicate, with the mean used. Lipids and glucose were determined through an overnight fasting (≥10 h) blood draw analyzed via a standard enzymatic colorimetric assay (c16000 Clinical Chemistry Analyzer, Abbott Diagnostics, Abbott Park, IL, USA). Lipid-lowering medication use (yes/no) was self-reported. Metabolic syndrome was classified using the harmonized definition [34] as outlined in Appendix A, and a unitless continuous metabolic syndrome risk score was calculated, consistent with previous scoring (lower values being desirable) [35,36]. Each of the five metabolic syndrome components were log10-transformed, then standardized as z-scores—(value − population mean)/SD and (population mean − value)/SD for HDL—then averaged to yield a final unitless score (see Appendix A). The z-scores used population means such that, for each biomarker, z > 0 indicates levels that are worse than average for the population of Australian women [37].

Quality of life was assessed using the Patient-Reported Outcome Measurement Information System (PROMIS) Global Health Scale, which solicits information across physical function, fatigue, pain, emotional distress, and social health, and provides summary scores for global physical and mental health components, with higher scores indicating better functioning [38]. Fatigue was assessed using the Functional Assessment of Chronic Illness Therapy Fatigue Scale, with higher scores indicating lower fatigue [39]. Arthralgia was measured using the Musculoskeletal Pain subscale from the Breast Cancer Prevention Trial Symptom Scale, with higher scores indicating worse pain [40]. Menopausal symptoms were assessed using the Greene Climacteric Scale [41]—psychological, somatic, and vasomotor symptoms subscales—with higher scores indicating more severe symptoms. Fear of cancer recurrence was measured using the Concerns About Recurrence Questionnaire, with higher scores indicating greater fear [42]. Body image was assessed using the Body Image and Relationships Scale (total score), with higher scores indicating greater impairment [43].

#### 2.4.3. Adverse Events

At each follow-up assessment, participants self-reported any adverse events (AEs), with severity categorized according to the Common Terminology Criteria for Adverse Events (CTC-AE; v4.0) from Grade 1 ‘mild’ to Grade 5 ‘fatal/death’. The ‘relatedness’ of the AE to the intervention was also recorded on a 5-point scale from ‘clearly not related’ to ‘clearly related’.

### 2.5. Sample Size

The sample size was calculated to provide at least 90% power (5% two-tailed significance) to detect a between-arm minimum difference of 5% body weight [9] and at least 80% power to detect effects of 0.5 SD in secondary outcomes [28].

### 2.6. Data Analysis

Multivariable linear mixed models were used to evaluate primary and secondary outcomes. Marginal means evaluated at mean values were used to report within-arm changes and between-arm differences (intervention effects). Transformed outcomes were back-transformed prior to reporting. Standardized intervention effect sizes were reported using Cohen’s *d* statistic. Based on a priori criteria [28], no potentially confounding variable met criteria for inclusion in models. Accordingly, models included fixed effects for the treatment arm, timepoint (6/12/18 months), and their interaction, along with the baseline value of the outcome [28]. To account for repeated measures from participants, models used restricted maximum likelihood estimation with an unstructured within-subjects covariance structure and no random intercept. The association between treatment arm and adverse events was assessed using Poisson regression.

Analyses followed intention-to-treat principles. Missing data were handled both using evaluable-case analysis and by multiple imputation (chained equations with m = 50 imputations) as sensitivity analyses since data were not missing completely at random. Variables included in imputation models are shown in Appendix A. Due to the potential influence of medication use (endocrine treatment, lipid-lowering, and blood pressure medications), further sensitivity analyses were performed that adjusted for baseline and concurrent use of these medications on related outcomes (metabolic syndrome risk score, HDL-cholesterol, triglycerides, blood pressure, musculoskeletal pain, and menopausal symptoms). Statistical significance was set at *p* < 0.05 (two-tailed). Analyses were performed in Stata v16 (StataCorp LLC, College Station, TX, USA).

## 3. Results

Of the 394 women contacted, 170 were ineligible, 65 declined to participate, and 159 women (71% of those eligible) consented and were randomized (Figure 1). Baseline characteristics were similar between arms (Table 1), with the only noteworthy differences (≥10%) being a greater proportion of post-menopausal and fewer peri-menopausal women at diagnosis, and a greater proportion with multi-comorbidities in the intervention versus usual care arm. Otherwise, women were, on average, 55 years old, approximately 11 months post-diagnosis, and half had obesity.

Retention was 89.9% at 6 months, 81.8% at 12 months, and 80.5% at 18 months, with 124 (78.0%) participants completing all four assessments. Drop-out differed by arm, with 13.9% (*n* = 11) in intervention versus 30.0% (*n* = 24) in usual care (*p* = 0.02). Relative to those completing all assessments, drop-outs were younger, reported lower physical quality of life, and were more likely to have children at home, lower income, non-English speaking background, and received both chemotherapy and radiotherapy (see Appendix A). Of the intervention participants, 73.4% (*n* = 58) received at least 75% (≥17 out of 22) of intended calls, defined a priori.

### 3.1. Weight and Body Composition

Significantly greater weight loss was observed in the intervention versus usual care arms at 12 months (−4.5% [95%CI: −6.5, −2.5], *p* < 0.001), which was largely maintained at 18 months (−3.1% [−5.3, −0.9], *p* = 0.007) (Table 2). Significant intervention effects on fat mass were observed at each assessment, with greater loss of lean mass observed in the intervention versus usual care at all follow-up assessments, being statistically significant at 6 and 18 months. Sensitivity analyses accounting for missing data (see Appendix A) led to similar intervention effects (±20%) and conclusions regarding clinical relevance and statistical significance. Analysis of regional body composition showed that across each region, fat mass decreased primarily with small decreases in lean mass, and small to no change in bone mass, leading to lower proportions of body fat, higher proportions of lean mass, and slightly higher or similar percentages of bone mass within each body region (see Appendix A).

### 3.2. Metabolic Syndrome

The intervention arm demonstrated statistically significant and more favorable metabolic syndrome risk scores across all follow-up assessments, which were statistically significant compared to usual care (Table 2). For individual metabolic syndrome components, significant intervention effects were observed for waist circumference at all follow-ups, systolic and diastolic blood pressure at 6 months only, and fasting plasma glucose at 12 months only. Sensitivity analyses adjusting for medication use (see Appendix A) and accounting for missing data (see Appendix A) yielded similar effect sizes and the same conclusions.

### 3.3. Patient-Reported Outcomes

Overall, the patient-reported outcomes (Table 3) favored intervention over usual care at most or all follow-up assessments. Significant intervention effects favoring intervention were seen at 12 months for physical quality of life (*d* = 0.40), musculoskeletal pain (*d* = −0.49), and body image (*d* = −0.31), with non-significant, small (*d* ≈ 0.2–0.3) improvements observed for mental quality of life and psychological menopausal symptoms. At 18 months, most effects were attenuated. Changes in endocrine treatment medications did not account for observed effects on musculoskeletal pain or menopausal symptoms (see Appendix A). After multiple imputation, effects on musculoskeletal pain were attenuated slightly and no longer significant at 12 months, while effects for quality of life and fatigue were of similar magnitude but no longer significant for physical quality of life at 6 months (see Appendix A). For the remaining patient-reported outcomes, the magnitude of effects changed slightly, but with no change to overall conclusions.

### 3.4. Adverse Events

Twenty-five serious AEs (SAE; CTC-AE grade 3–5) from 21 participants were observed over the trial (intervention: *n* = 13; usual care: *n* = 12) (Table 4). Only two of the SAEs were considered possibly related to the intervention (knee and foot injuries), neither of which was permanently disabling or life-threatening. Additionally, 180 moderate (grade 2) AEs were reported (intervention: *n* = 96 events, 53 participants; usual care: *n* = 84 events, 40 participants)—of these, 18 in the intervention arm were considered possibly related to the intervention and one was considered probably related, and all were primarily musculoskeletal injuries. There were no significant between-arm differences in the rate of either serious AEs (incidence rate ratio; IRR = 0.98 [95%CI: 0.41, 2.34], *p* = 0.95) or moderate AEs (IRR = 1.03 [95%CI: 0.76, 1.40], *p* = 0.85).

## 4. Discussion

Intervention participants achieved statistically significant and clinically meaningful weight loss and improvement in metabolic syndrome risk at 12 months compared with usual care participants. Importantly, these improvements were largely sustained six months after intervention contact ceased, highlighting the durable effects of the intervention. Further, beneficial effects on patient-reported outcomes were observed. The magnitude of weight loss achieved is comparable to that observed in previous weight loss trials in breast cancer survivors [44,45,46,47], with the intervention effect on weight observed (−4.5% [−6.5, −2.5]) encompassing the clinically meaningful difference of 5% weight loss [9]. These findings provide further support for the use of remotely delivered interventions to successfully achieve weight loss [44,46,47], as well as the feasibility and acceptability of offering such interventions soon after diagnosis and treatment completion.

At study baseline, almost 50% of women were classified as having metabolic syndrome, putting them at considerably increased health risk [26,27]. Those allocated to the intervention observed significant and sustained improvements in metabolic syndrome risk score, with an effect size (Cohen’s *d*) of approximately −0.3. This effect on metabolic syndrome risk is smaller than that observed by Dieli-Conwright et al. [24]; however, the baseline prevalence of metabolic syndrome (77%) was considerably higher in this previous trial of highly sedentary and largely Hispanic breast cancer survivors—in addition, the trial evaluated an intensive supervised exercise intervention. The outcomes observed in the present trial likely reflect the more realistic magnitude of effect achievable with a scalable, telehealth intervention.

Notably, though, we did not observe any significant or clinically meaningful intervention effects on lipids or blood pressure at the end of the intervention. Previous exercise-only intervention trials in breast cancer survivors with good adherence to exercise prescription have demonstrated small–large effects on triglycerides and HDL-cholesterol [24,48], but not in trials with lower adherence [49,50]. Dietary intervention studies have reported differing effects on lipids depending on the macronutrient composition of the diet—significant improvements in triglycerides with the low-carbohydrate diet group only, and a significant, albeit small, improvement in HDL-cholesterol in the low-fat group only [51]. A more specific focus on macronutrient composition or dietary patterns and more frequent assessment of adherence to exercise and dietary prescriptions may be necessary to improve lipid profiles and ultimately metabolic risk in breast cancer survivors. The Mediterranean diet is a dietary pattern that has shown consistent beneficial metabolic and cardiovascular effects in a number of populations [52,53]; however, there has been limited investigation of its benefits in breast cancer survivors [11]. Given the large burden of cardiovascular mortality in breast cancer survivors [17], the benefit of a Mediterranean-style diet and exercise intervention warrants further investigation.

Recent evidence suggests that body composition, defined by low muscle mass (sarcopenia) and adiposity, is more strongly associated with poorer survival in breast cancer than BMI [16]. In this trial, a significant reduction in total fat mass and central adiposity was observed with the intervention; however, reductions in lean mass of ≈1 kg were also observed, consistent with what is typically observed with weight loss [54]. When examined across body region, loss of lean mass occurred in every region, though not necessarily to equal degrees. Being much less than the loss of fat mass, the relative body composition shifted towards a higher percentage of lean mass. Although resistance exercise was encouraged, many women chose a less intensive resistance exercise program. Interventions emphasising supervised progressive resistance training, and perhaps gym-based sessions using specialized equipment [55,56], may be needed to minimize muscle loss. Further evidence on how to effectively achieve similar benefits via remotely delivered interventions, such as with telehealth, is needed [57].

This trial also examined the effect of the intervention on key patient-reported outcomes, including quality of life, and treatment-related side effects. A significant intervention effect on physical quality of life was observed, similar to improvements observed in previous trials [44,45], where significant short-term intervention effects were observed [44]. However, a particularly novel and important finding is the significant medium–large intervention effects observed for musculoskeletal pain, which was used to assess arthralgia. Arthralgia is common in breast cancer survivors, particularly those treated with aromatase inhibitors (AIs) [19], and can often lead to poor adherence or discontinuation of AI treatment [58,59,60]. Recent studies of exercise interventions have shown improvements in arthralgia and pain scores following intervention [61,62]; however, these trials exclusively recruited women on AIs reporting arthralgia/joint pain. Given the magnitude of intervention effects observed in our sample, where only a third were treated with AIs, and the very limited evidence to date [63], this finding warrants further investigation, as does the potential beneficial effect on menopausal symptoms (neither of which were attenuated following adjustment for changes in endocrine treatment).

Several ongoing trials are evaluating the effect of weight loss interventions on breast cancer-specific outcomes, including survival [13,14,15]. Preliminary findings from the SUCCESS-C trial [64] showed no significant effect on disease-free survival in the lifestyle intervention arm (vs. non-lifestyle intervention) in intention-to-treat analyses; however, weight loss in their telephone-delivered lifestyle arm at two-year follow-up was very modest (mean: 1.0 kg) and attrition was exceptionally high (51.8%) [64]. Post-hoc analyses in lifestyle intervention arm completers (vs. non-intervention) suggest a significant benefit for disease-free survival (HR: 0.51 [95%CI: 0.33, 0.78]) [64]. These results show promise but highlight the challenges of achieving and maintaining clinically meaningful weight loss (≥5%) and participant retention.

Strengths of this trial include the evaluation of a remotely delivered intervention, with potential for wider implementation; assessment of the durability of intervention effects; and the inclusion and retention of a broadly representative sample of breast cancer survivors. However, the sample included an over-representation of younger breast cancer survivors, with almost 40% reporting being pre-menopausal at diagnosis. Given that 22% of breast cancer cases are diagnosed in women <50 years in Australia [65], this likely reflects a particular interest and need for such interventions among younger survivors, who have been excluded from many of the previous trials [11,12]. Limitations of the trial include the primarily Caucasian sample, which limits generalizability to non-Caucasian populations. Although there was differential attrition, multiple imputation models showed that this did not affect the main study findings. The study was sufficiently powered to detect clinically important differences in the primary outcome and effect sizes in secondary outcomes of 0.5 SD—for some secondary outcomes, smaller differences were observed, which may still be clinically meaningful, but for which we were underpowered to detect between-arm effects. These should be examined in future trials or via pooling of trial findings in meta-analyses.

## 5. Conclusions

The COVID-19 pandemic has highlighted the need for quality cancer care that can be remotely delivered—an already advanced area of research in the field of cancer survivorship and lifestyle intervention. This trial adds to this evidence as both clinically meaningful and durable weight loss and improvement in metabolic syndrome risk were achieved, in women following a breast cancer diagnosis. Future research should further explore strategies for maximizing the health benefits achievable with remotely delivered interventions, particularly in relation to minimizing loss in lean mass, improvements in arthralgia and menopausal symptoms, and for achieving improvements across all metabolic syndrome components.

## Figures and Tables

**Figure 1 nutrients-13-04091-f001:**
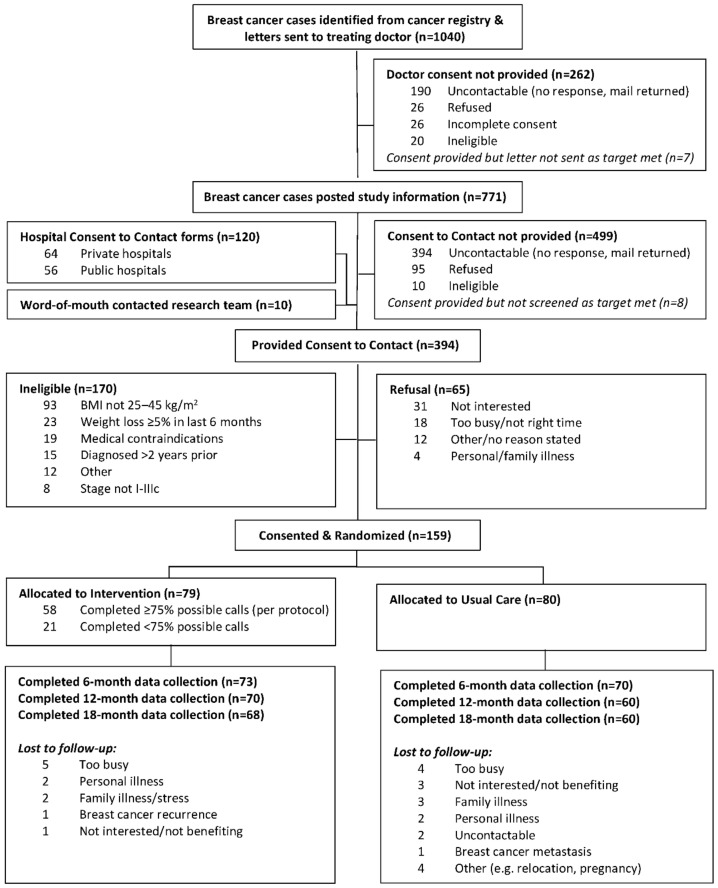
Participant flow diagram for Living Well after Breast Cancer trial.

**Table 1 nutrients-13-04091-t001:** Baseline participant characteristics in the Living Well after Breast Cancer trial (*n* = 159).

Characteristic	Usual Care (*n* = 80)	Intervention (*n* = 79)
	Mean (SD)	
Age (years)	54.9 (9.3)	55.9 (9.1)
BMI (kg/m^2^)	31.3 (5.2)	31.4 (4.9)
Months since diagnosis	10.8 (5.3)	10.7 (4.8)
Months since treatment completion	4.9 (4.6)	5.2 (4.7)
	*n* (%)	
Menopausal status at diagnosis		
Premenopausal	31 (39%)	28 (35%)
Perimenopausal ^a^	15 (19%)	6 (8%)
Postmenopausal ^a^	34 (42%)	45 (57%)
Breast cancer stage ^b^		
Stage 1	46 (58%)	40 (51%)
Stage 2	24 (30%)	30 (38%)
Stage 3	9 (11%)	9 (11%)
Estrogen receptor status ^b^		
Positive	72 (91%)	67 (85%)
Negative	7 (9%)	12 (15%)
HER2 ^b^		
Positive	9 (11%)	11 (14%)
Negative	68 (86%)	68 (86%)
Equivocal	2 (3%)	0 (0%)
Chemotherapy treatment	51 (64%)	48 (61%)
Radiotherapy treatment	63 (79%)	63 (80%)
Endocrine treatment		
None	35 (44%)	32 (41%)
SERM	19 (24%)	22 (28%)
Aromatase inhibitor	26 (32%)	24 (30%)
GnRH agonist	0 (0%)	1 (1%)
Metabolic syndrome present	37 (48%)	37 (47%)
Charlson Comorbidity Index ^c^		
0	51 (64%)	47 (60%)
1	18 (22%)	13 (16%)
≥2 ^a^	11 (14%)	19 (24%)
Married or stable union	56 (70.0%)	54 (68.4%)
Caucasian	78 (97.5%)	78 (98.7%)
Employment status		
Paid work	44 (55%)	50 (63%)
Retired, home duties, unable to work, other	36 (45%)	29 (37%)
	*n* (%)	
Highest education level		
High school or less	32 (40%)	32 (41%)
Technical/trade/diploma	21 (26%)	16 (20%)
University or higher	27 (34%)	31 (39%)
Gross household income (AUD) ^d^		
<$82,056 per year	34 (42%)	37 (47%)
≥$82,056 per year	38 (48%)	33 (42%)
Not reported/not known	8 (10%)	9 (11%)

Abbreviations: AUD, Australian dollar; BMI, body mass index; GnRH, gonadotropin releasing hormone; HER2, human epidermal growth receptor 2; SERM, selective estrogen receptor modulators. ^a^ Noteworthy difference (≥10%) between arms. ^b^ Percentages exclude missing data (*n* = 1, usual care arm). ^c^ Charlson Comorbidity Index was based on self-reported diagnosis of 13 conditions [17], with the addition of hypertension. ^d^ Threshold indicates 60th percentile of Australian population household income based on 2007–2008 census.

**Table 2 nutrients-13-04091-t002:** Within-arm and between-arm changes for weight, body composition, and metabolic syndrome risk biomarkers: Living Well after Breast Cancer trial.

Outcome	Timepoint	Intervention		Usual Care		Intervention Effect (Intervention—Usual Care)		
		*n*	Mean Change (95% CI)	*n*	Mean Change (95% CI)	Mean Difference (95% CI)	*p*	d ^a^
Weight (% of baseline value)	Baseline M (SD)	79	83.9 (14.2)	80	83.6 (13.6)			
	6 months	73	−4.61 (−5.77, −3.44)	70	−0.52 (−1.70, 0.67)	−4.09 (−5.75, −2.43)	<0.001	−0.30
	12 months ^b^	70	−5.06 (−6.46, −3.66)	60	−0.58 (−2.02, 0.85)	−4.48 (−6.48, −2.47)	<0.001	−0.32
	18 months	68	−3.69 (−5.23, −2.16)	60	−0.62 (−2.22, 0.97)	−3.07 (−5.28, −0.86)	0.007	−0.22
Weight (kg)	Baseline M (SD)	79	83.9 (14.2)	80	83.6 (13.6)			
	6 months	73	−3.74 (−4.71, −2.76)	70	−0.43 (−1.42, 0.56)	−3.31 (−4.70, −1.92)	<0.001	−0.24
	12 months ^b^	70	−4.12 (−5.28, −2.96)	60	−0.52 (−1.71, 0.67)	−3.60 (−5.26, −1.94)	<0.001	−0.26
	18 months	68	−3.03 (−4.34, −1.73)	60	−0.56 (−1.91, 0.80)	−2.48 (−4.36, −0.59)	0.010	−0.18
Total fat mass (kg)	Baseline M (SD)	73	38.8 (10.4)	70	37.5 (10.2)			
	6 months	67	−3.13 (−3.97, −2.30)	62	0.13 (−0.73, 1.00)	−3.26 (−4.47, −2.06)	<0.001	−0.32
	12 months ^b^	64	−3.27 (−4.26, −2.29)	54	0.05 (−0.98, 1.08)	−3.32 (−4.75, −1.90)	<0.001	−0.32
	18 months	63	−2.11 (−3.19, −1.03)	54	−0.29 (−1.43, 0.85)	−1.82 (−3.39, −0.25)	0.023	−0.18
Total lean mass (kg)	Baseline M (SD)	73	42.8 (5.0)	70	43.6 (5.2)			
	6 months	67	−0.96 (−1.28, −0.63)	62	−0.24 (−0.57, 0.09)	−0.71 (−1.18, −0.25)	0.002	−0.14
	12 months ^b^	64	−1.07 (−1.46, −0.68)	54	−0.52 (−0.93, −0.10)	−0.55 (−1.12, 0.02)	0.059	−0.11
	18 months	63	−1.20 (−1.63, −0.77)	54	−0.14 (−0.59, 0.32)	−1.06 (−1.68, −0.43)	<0.001	−0.21
Metabolic syndrome risk score	Baseline M (SD)	78	0.65 (0.60)	77	0.63 (0.59)			
	6 months	69	−0.19 (−0.27, −0.11)	65	0.03 (−0.05, 0.12)	−0.22 (−0.34, −0.10)	<0.001	−0.37
	12 months ^b^	67	−0.18 (−0.27, −0.08)	56	0.01 (−0.09, 0.11)	−0.19 (−0.32, −0.05)	0.006	−0.32
	18 months	66	−0.15 (−0.24, −0.06)	57	0.01 (−0.08, 0.11)	−0.16 (−0.29, −0.03)	0.014	−0.27
Waist circumference (cm)	Baseline M (SD)	79	106.7 (11.7)	80	104.9 (10.4)			
	6 months	73	−3.47 (−4.95, −1.99)	70	−0.64 (−2.14, 0.87)	−2.83 (−4.94, −0.71)	0.009	−0.26
	12 months ^b^	70	−5.50 (−7.11, −3.89)	60	−2.30 (−3.98, −0.62)	−3.20 (−5.53, −0.87)	0.007	−0.29
	18 months	68	−5.29 (−6.81, −3.78)	60	−2.50 (−4.08, −0.91)	−2.80 (−4.99, −0.61)	0.012	−0.25
Triglycerides (mmol/L) ^c^	Baseline M (SD)	78	1.4 (0.7)	78	1.5 (0.9)			
	6 months	71	−0.03 (−0.12, 0.05)	67	0.08 (−0.01, 0.18)	−0.11 (−0.24, 0.01)	0.081	−0.14
	12 months ^b^	67	−0.08 (−0.18, 0.02)	57	0.04 (−0.08, 0.16)	−0.12 (−0.28, 0.03)	0.125	−0.15
	18 months	66	−0.11 (−0.20, −0.02)	59	−0.01 (−0.11, 0.09)	−0.10 (−0.24, 0.03)	0.124	−0.13
HDL-cholesterol (mmol/L)	Baseline M (SD)	78	1.4 (0.3)	78	1.4 (0.4)			
	6 months	71	0.02 (−0.02, 0.07)	67	−0.02 (−0.06, 0.03)	0.04 (−0.02, 0.10)	0.182	0.13
	12 months ^b^	67	0.05 (0.00, 0.09)	57	−0.01 (−0.06, 0.04)	0.06 (−0.01, 0.12)	0.110	0.17
	18 months	66	0.06 (0.01, 0.11)	59	0.00 (−0.04, 0.05)	0.06 (−0.01, 0.12)	0.097	0.17
Systolic blood pressure (mmHg)	Baseline M (SD)	79	125.3 (12.2)	79	123.4 (11.3)			
	6 months	71	−1.69 (−4.21, 0.83)	68	3.44 (0.86, 6.02)	−5.13 (−8.73, −1.52)	0.005	−0.44
	12 months ^b^	70	1.05 (−1.70, 3.80)	59	2.20 (−0.77, 5.17)	−1.15 (−5.20, 2.90)	0.577	−0.10
	18 months	68	3.07 (−0.37, 6.52)	59	5.54 (1.88, 9.19)	−2.46 (−7.48, 2.56)	0.336	−0.21
Diastolic blood pressure (mmHg)	Baseline M (SD)	79	78.7 (9.4)	79	77.9 (7.3)			
	6 months	71	−0.35 (−2.03, 1.32)	68	2.40 (0.69, 4.12)	−2.76 (−5.15, −0.36)	0.024	−0.33
	12 months ^b^	70	0.60 (−1.07, 2.27)	59	1.51 (−0.29, 3.30)	−0.90 (−3.36, 1.55)	0.470	−0.11
	18 months	68	1.06 (−0.80, 2.92)	59	3.39 (1.41, 5.36)	−2.33 (−5.04, 0.39)	0.093	−0.28
Fasting plasma glucose (mmol/L)	Baseline M (SD)	78	5.5 (1.2)	78	5.6 (1.1)			
	6 months	71	−0.34 (−0.49, −0.19)	67	−0.21 (−0.36, −0.05)	−0.13 (−0.35, 0.08)	0.230	−0.11
	12 months ^b^	67	−0.17 (−0.32, −0.03)	57	0.06 (−0.10, 0.21)	−0.23 (−0.44, −0.02)	0.032	−0.20
	18 months	66	−0.12 (−0.29, 0.04)	59	−0.10 (−0.27, 0.08)	−0.02 (−0.26, 0.22)	0.844	−0.02

Abbreviations: HDL, high-density lipoprotein. ^a^ Standardized effect: mean intervention effect divided by pooled baseline standard deviation of the outcome. ^b^ End-of-intervention contact; primary endpoint. ^c^ Modeled as log outcome adjusted for log outcome at baseline, with results back-transformed to change (follow-up minus baseline) in original units using the relevant expression of marginal means.

**Table 3 nutrients-13-04091-t003:** Within-arm and between-arm changes for patient-reported outcomes: Living Well after Breast Cancer trial.

Outcome	Timepoint	Intervention		Usual Care		Intervention Effect (Intervention—Usual Care)		
		*n*	Mean Change (95% CI)	*n*	Mean Change (95% CI)	Mean Difference (95% CI)	*p*	d ^a^
QOL Physical Health component (T score) ^b^	Baseline M (SD)	78	44.8 (6.9)	77	45.9 (6.6)			
	6 months	69	2.43 (1.31, 3.56)	65	0.46 (−0.70, 1.62)	1.98 (0.36, 3.59)	0.017	0.29
	12 months ^c^	65	3.16 (1.82, 4.50)	58	0.50 (−0.90, 1.91)	2.66 (0.71, 4.60)	0.007	0.39
	18 months	65	1.56 (0.13, 2.99)	57	0.38 (−1.13, 1.89)	1.18 (−0.90, 3.26)	0.266	0.18
QOL Mental Health component (T score) ^b^	Baseline M (SD)	78	46.1 (7.2)	77	45.5 (6.3)			
	6 months	69	1.34 (0.10, 2.57)	65	−0.17 (−1.44, 1.10)	1.51 (−0.27, 3.28)	0.097	0.22
	12 months ^c^	65	1.98 (0.68, 3.28)	58	0.21 (−1.15, 1.58)	1.77 (−0.12, 3.66)	0.067	0.26
	18 months	65	−0.36 (−1.98, 1.26)	57	0.17 (−1.54, 1.88)	−0.53 (−2.89, 1.83)	0.659	−0.08
Fatigue ^b^	Baseline M (SD)	77	35.5 (9.7)	77	37.6 (9.5)			
	6 months	68	3.21 (1.57, 4.86)	65	0.75 (−0.94, 2.43)	2.47 (0.11, 4.83)	0.040	0.26
	12 months ^c^	64	4.29 (2.57, 6.01)	58	2.26 (0.46, 4.05)	2.03 (−0.46, 4.53)	0.110	0.21
	18 months	63	2.63 (0.81, 4.46)	57	1.99 (0.09, 3.89)	0.65 (−2.00, 3.29)	0.632	0.07
Musculoskeletal Pain	Baseline M (SD)	63	1.5 (1.1)	59	1.6 (1.0)			
	6 months	56	−0.21 (−0.44, 0.02)	50	0.40 (0.16, 0.64)	−0.61 (−0.94, −0.28)	<0.001	−0.58
	12 months ^c^	52	−0.19 (−0.42, 0.04)	46	0.32 (0.08, 0.56)	−0.51 (−0.84, −0.18)	0.003	−0.49
	18 months	53	−0.07 (−0.32, 0.18)	44	0.26 (−0.01, 0.52)	−0.33 (−0.69, 0.04)	0.079	−0.31
Menopausal Symptoms—Psychological subscale	Baseline M (SD)	75	10.0 (6.2)	76	9.6 (5.7)			
	6 months	66	−1.40 (−2.51, −0.30)	65	−0.14 (−1.25, 0.98)	−1.27 (−2.84, 0.30)	0.113	−0.21
	12 months ^c^	62	−1.90 (−3.20, −0.60)	58	−0.62 (−1.96, 0.71)	−1.28 (−3.14, 0.59)	0.179	−0.22
	18 months	61	−1.24 (−2.54, 0.06)	56	−0.22 (−1.56, 1.12)	−1.02 (−2.88, 0.85)	0.286	−0.17
Menopausal Symptoms—Somatic subscale	Baseline M (SD)	76	5.5 (4.3)	76	5.1 (4.0)			
	6 months	67	−0.67 (−1.31, −0.04)	65	0.58 (−0.07, 1.22)	−1.25 (−2.15, −0.34)	0.007	−0.30
	12 months ^c^	62	−0.77 (−1.53, −0.01)	58	−0.07 (−0.85, 0.71)	−0.70 (−1.79, 0.39)	0.206	−0.17
	18 months	63	−0.67 (−1.47, 0.13)	56	0.27 (−0.56, 1.10)	−0.94 (−2.10, 0.21)	0.108	−0.23
Menopausal Symptoms—Vasomotor subscale	Baseline M (SD)	76	2.6 (2.2)	76	2.4 (2.1)			
	6 months	67	0.24 (−0.13, 0.62)	65	0.49 (0.11, 0.87)	−0.25 (−0.78, 0.29)	0.367	−0.12
	12 months ^c^	63	0.06 (−0.36, 0.48)	58	0.42 (−0.02, 0.85)	−0.35 (−0.96, 0.25)	0.250	−0.17
	18 months	63	−0.22 (−0.65, 0.21)	56	0.32 (−0.13, 0.77)	−0.54 (−1.16, 0.08)	0.089	−0.25
Fear of Cancer Recurrence	Baseline M (SD)	77	14.5 (9.6)	76	15.5 (9.7)			
	6 months	68	−2.20 (−3.74, −0.65)	64	−1.08 (−2.67, 0.51)	−1.12 (−3.34, 1.10)	0.321	−0.12
	12 months ^c^	64	−2.24 (−3.83, −0.65)	57	−3.35 (−5.02, −1.67)	1.11 (−1.21, 3.42)	0.348	0.12
	18 months	64	−2.45 (−4.24, −0.65)	55	−1.99 (−3.91, −0.07)	−0.46 (−3.08, 2.17)	0.734	−0.05
Body Image—Total score	Baseline M (SD)	78	2.8 (0.6)	77	2.7 (0.6)			
	6 months	69	−0.35 (−0.46, −0.24)	65	−0.14 (−0.25, −0.03)	−0.21 (−0.37, −0.05)	0.010	−0.36
	12 months ^c^	65	−0.43 (−0.54, −0.32)	58	−0.25 (−0.36, −0.13)	−0.18 (−0.35, −0.02)	0.030	−0.31
	18 months	65	−0.30 (−0.42, −0.17)	57	−0.21 (−0.35, −0.08)	−0.08 (−0.27, 0.10)	0.380	−0.14

Abbreviations: QOL, quality of life. ^a^ Standardized effect: mean intervention effect divided by pooled baseline standard deviation of the outcome. ^b^ Higher scores are preferable. ^c^ End-of-intervention contact; primary endpoint.

**Table 4 nutrients-13-04091-t004:** Serious adverse events reported within each arm over the 18-month study period: Living Well after Breast Cancer trial.

	Intervention (*n* = 11 Participants)	No. of Events	Usual Care (*n* = 10 Participants)	No. of Events
Life-threatening (*n* = 4) ^a^	Stage IV breast cancer (bone metastasis)	1	Heart episode during surgery	1
			Stage IV breast cancer (i.e., bone metastasis, site unknown)	2
Severe/undesirable (*n* = 21) ^b^	Musculoskeletal events requiring hospitalization or surgery	6 ^c^	Musculoskeletal events requiring hospitalization or surgery	1
	Genitourinary events requiring hospitalization or surgery	4	Gastrointestinal events requiring hospitalization or surgery	1
	Other events requiring hospitalization or surgery	1	Genitourinary events requiring hospitalization or surgery	2
	Local breast cancer recurrence	1	Respiratory events requiring hospitalization or surgery	3
			Other events requiring hospitalization or surgery	2

^a^ Life-threatening symptoms. ^b^ Significant symptoms requiring hospitalization or invasive intervention. ^c^ Includes two adverse events possibly related to the intervention (i.e., knee injury, *n* = 1; and foot injury, *n* = 1).

## Data Availability

Data available upon request to corresponding author with ethics approval.

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
