# Peer review of "Effect of a Remotely Delivered Weight Loss Intervention in Early-Stage Breast Cancer: Randomized Controlled Trial"

_nutrients, 2021, doi:10.3390/nu13114091_

Round 1

Reviewer 1 Report

This intervention trial is important and suggest the possibility to follow breast cancer women and modify their lifestyle  by using remotely-delivered intervention. I would suggest to improve the discussion session pointing potential solutions to the loss of lean mass. Authors discussed the loss of lean mass that is associated with weight loss. However, I would suggest to take a look to TANITA data on muscle mass (in specific area on the body if is possible) and not only the global lean mass. Women in fact reduced continuately their lean mass also at 12 and 18 months while the fat mass substantially remained constant on average.  Consistently we did not observe significant improvement in HDL cholesterol and blood pressure that usually accompain physical exercise. I would suggest to comment these points.

Author Response

Body composition was measured in this trial via DXA (Lunar Prodigy), which provides estimates for regional body composition in addition to total body (android, gynoid, trunk, legs, arms). We have included the analysis of regional body composition as exploratory analyses, with the results included as supplementary tables. The findings from this analysis supported the overall findings from the total body composition, in that weight change was primarily from loss of fat mass and small loss of lean mass across all body regions, but to slightly differing degrees. Amendments have been made to the methods, results, and discussion section to reflect the addition of this exploratory analysis. However, we don’t believe these findings add any more to or change our  recommendations provided in the discussion on suggestions for reducing loss of lean mass.

We have added a paragraph to the Discussion section where we have commented on our lack of intervention effects on lipids and blood pressure. All changes are highlighted as track changes in the manuscript. 

Reviewer 2 Report

Thank you for your contribution!

This study is well designed, long term-prospective study to compare between the remotely-delivered intervention and usual care in breast cancer survivors.

Even though there showed some fat free mass decrease, this study results are very important for general population with obese breast cancer survivors, who should be very cautious not gaining body weight. Muscle mass loss is inevitable in some sense when reduction of body weight even in general obese subjects. Therefore, proper nutrition and exercise program not to lose muscle mass should be considered. 

Author Response

We thank the Reviewer for their encouraging comments. We believe the additions made to the manuscript based on Reviewer 1’s suggestions also help to acknowledge the point made by Reviewer 2.